# Microstructure and Key Properties of Phosphogypsum-Red Mud-Slag Composite Cementitious Materials

**DOI:** 10.3390/ma15176096

**Published:** 2022-09-02

**Authors:** Feiyue Ma, Liangliang Chen, Zhiwei Lin, Zhuo Liu, Weichuan Zhang, Rongxin Guo

**Affiliations:** 1Yunnan Key Laboratory of Disaster Reduction in Civil Engineering, Faculty of Civil Engineering and Mechanics, Kunming University of Science and Technology, Kunming 650500, China; 2Broadvision Engineering Consultants, No. 9 Shuangfeng Rd., Kunming 650299, China

**Keywords:** phosphogypsum, red mud, slag, hydraulic cementitious material, hydration product, strength

## Abstract

Due to the low content of silicon and aluminum in red mud and the low reaction activity of red mud, when it was used to prepare composite cementitious materials, it was necessary to assist other aluminosilicates and improve their activity by certain methods. In this study, it was proposed to add slag to increase the percentage of silicon and aluminum in the system, and to improve the reactivity of the system through the activation effect of sulfate in phosphogypsum. The effects of slag and phosphogypsum contents on the mechanical properties and microstructures of composite cementitious materials were studied. X-ray diffraction analysis (XRD), thermogravimetric analysis (TG-DTG), and scanning electron microscopy (SEM) were used to analyze the effects of slag and phosphogypsum contents on the hydration products, microstructure, and strength formation mechanism of composite cementitious materials. The results show that with the increase of slag, the strength of the composite cementitious material increases gradually. When the slag content is 50%, the 28-day compressive strength reaches a maximum of about 14 MPa. Compared with the composite material without phosphogypsum, the composite cementitious material with 10–20% phosphogypsum showed higher strength properties, in which the 28-day compressive strength exceeds 24 MPa. The main reason for this is that the sulfate in phosphogypsum can cause the composite cementitious material to generate a large amount of ettringite and accelerate the dissolution of red mud and slag, increasing the release of aluminates, silicates, and Ca^2+^ to form more C-(A)-S-H and ettringite. In addition, a large amount of C-(A)-S-H makes ettringite and unreacted particles combine into a uniform and compact structure, thus improving the strength. When the content of phosphogypsum exceeds 40%, the 28-day compressive strength of the composite cementitious material drops below 12 MPa due to the presence of fewer hydration products and the expansion of ettringite.

## 1. Introduction

Phosphogypsum is a solid waste discharged during the wet production of phosphoric acid by phosphorus chemical companies. Its main component is CaSO_4_·2H_2_O, and it also contains a small number of impurities, such as soluble phosphorus (P_2_O_5_), fluorine (F^−^), organic substances, etc.; its pH value is 2–3 [1,2]. It was reported that about 5 tons of phosphogypsum are produced for every ton of phosphoric acid produced. The global annual output of phosphogypsum is about 100–280 million tons, but the recycling rate is only 15% [3]. At present, the annual emission of phosphogypsum in China exceeds 75 million tons, and the cumulative stockpile has exceeded 500 million tons, but the comprehensive utilization rate is only about 40% [3]. Red mud is a reddish brown alkaline industrial waste discharged during the process of refining alumina from bauxite [4]; its main components are Fe_2_O_3_, Al_2_O_3_, SiO_2_, NaO_2_, CaO, TiO_2_, etc., and the pH value is 10–13 [5]. According to the alumina refining process, red mud can be divided into sintering process red mud, Bayer process red mud, and combined process red mud, of which Bayer process mud is the most widely used [5,6]. Depending on the type of bauxite, 1–1.5 tons of red mud is produced for every 1 ton of alumina produced. At present, the global reserves of red mud have reached 4.6 billion tons, of which China has accumulated more than 790 million tons. However, the comprehensive utilization rate is only 5.24% [5,7,8]. Therefore, how to improve the resource utilization of these two solid wastes is an urgent issue.

In some studies, phosphogypsum-based composite cementitious materials have been prepared by mixing phosphogypsum with latent hydraulic industrial waste residues and alkali activators [9]. According to the report, the preparation of phosphogypsum-based composite cementitious materials with untreated phosphogypsum has high later strength and excellent water resistance. However, the soluble phosphorus and fluorine in phosphogypsum will hinder the reaction of cement and slag or be converted into Ca_3_(PO_4_)_2_ and CaF_2_ covering the surface of phosphogypsum, hindering the dissolution of phosphogypsum, thereby reducing the early strength development of composite cementitious materials [10,11]. In order to weaken the influence of phosphogypsum impurities, high-temperature calcination, alkali neutralization, and water washing are mainly used to pretreat phosphogypsum. Compared with simple water washing and lime neutralization, the effect of calcined phosphogypsum is better [1]. However, the more complicated pretreatment methods can easily cause secondary pollution and are not conducive to large-scale production and ecological protection. Interestingly, Lin et al. [12] studied the effect of phosphogypsum on the properties of recycled cement. The study found that phosphogypsum can accelerate the formation of C-S-H and ettringite to improve the strength of recycled cement. Moreover, the impurities in phosphogypsum also participate in the reaction to generate calcium phosphate silicate.

Moreover, with the large-scale stockpiling of red mud, the resource utilization of red mud has also attracted wide attention. In recent years, much research has been conducted on red mud-based geopolymers. In most cases, red mud has been mainly used in combination with other aluminosilicates such as fly ash, slag, and silica fume [13]. Studies have shown that the reaction rate of red mud-based geopolymers depends on the dissolved NaOH concentration in red mud, and the mechanical properties were affected by the dissolved SiO_2_ and AlO_2_ in the geopolymers [14,15]. A study by Ye et al. [16] indicated that the soluble sodium aluminosilicate formed in alkali-heat-activated red mud can form an alkaline environment and promote the geopolymerization of aluminosilicates. From the current research situation, if the Bayer process red mud is only considered as the alkaline activator in the cementitious material, its utilization amount will be small, and the advantage of being rich in alumina and silica will not be fully considered. However, due to the low activity of red mud, the amount of red mud involved in the formation of geopolymers is limited [17,18]. Therefore, red mud must be activated when used in the field of cementitious materials.

In summary, the application of phosphogypsum and red mud in the field of building materials has been extensively studied. However, there have been few reports on the combined use of phosphogypsum and red mud. Phosphogypsum-based cementitious material mainly rely on the sulfate and alkaline activator activate the slag activity together to generate a large amount of ettringite and C-S-H gel, thereby generating strength. However, red mud contains a large amount of CaO, SiO_2_, Al_2_O_3_, and Na_2_O, and has a specific hydration activity and high alkaline properties [5], enabling it to fully meet the hardening conditions required for phosphogypsum-based cementitious materials. However, due to the low percentage of silica and alumina in red mud, other aluminosilicates must be added when red mud is used as the main raw material for the preparation of geopolymers [19,20]. Therefore, to further ensure the properties of composite cementitious materials, slag was added to increase the percentage of silica and alumina in this study. Using phosphogypsum, red mud, and slag as the main raw materials to prepare composite cementitious materials, the high alkalinity of red mud can be used to stimulate the potential activity of slag and neutralize the acidic impurities of phosphogypsum, thereby realizing the activation of slag and the pretreatment of phosphogypsum. In addition, the sulfate in gypsum can accelerate the geopolymerization of slag and red mud to improve the mechanical properties of red mud-based geopolymers and promote the synergistic utilization of slag, red mud, and gypsum [21].

In this study, the feasibility of preparing phosphogypsum-red mud-slag composite cementitious materials with Bayer process red mud, phosphogypsum, and slag as the main raw materials and adding 10% cement as the curing agent was explored. The influence of different phosphogypsum and slag contents on the strength of composite cementitious materials was studied, and the influence tendency of phosphogypsum and slag on the strength of phosphogypsum-red mud-slag composite cementitious materials was clarified. XRD, TG-DTG, and SEM were used to analyze the hydration products and microstructure of the composite cementitious material, and the influence of the mechanism is discussed.

## 2. Materials and Methods

### 2.1. Materails

The chemical components of phosphogypsum, red mud, slag, and cement were measured using an X-ray fluorescence spectrometer (XRF). The results are shown in Table 1. Phosphogypsum is a solid powder that was taken from Kunming City in Yunnan Province. It has a water content of 15–20% and a pH of 4.2. The phosphogypsum was dried to constant weight in an oven at 50 °C before the experiment. The specific surface area was determined to be 180 m^2^/kg by the Blaine method [22]. The main chemical components are SO_3_, CaO, and SiO_2_, and it also contains a small amount of soluble phosphorus (P_2_O_5_) and fluorine (F^−^). The mineral composition of phosphogypsum measured by XRD is mainly CaSO_4_·2H_2_O and SiO_2_ (Figure 1). SiO_2_ is one of the main sources of the potential activity of pozzolanic reactive substances such as slag and fly ash. It can be seen that phosphogypsum is not only the main source of sulfate, but also one of the sources of SiO_2_. The Bayer process red mud is a strongly alkaline industrial waste residue with a pH of 11.5, obtained from Wenshan Prefecture, Yunnan Province. The red mud with a high water content was dried to constant weight in an oven at 100 °C before the experiment. The main chemical components of red mud are Fe_2_O_3_, CaO, SiO_2_, Al_2_O_3_, and Na_2_O, among which Fe_2_O_3_ content is the highest, and the CaO, SiO_2_, and Al_2_O_3_ contents exceed 50%. This shows that it is feasible to use red mud as a raw material to prepare cementitious materials [23]. The main mineral components of red mud are cancrinite, calcite, dicalcium silicate, tricalcium aluminate, hematite, goethite, boehmite, gibbsite, etc. (Figure 1). As regards the slag (S95 grade), the specific surface area measured by the Blaine method [22] was 480 m^2^/kg, and it was taken from Qujing City, Yunnan Province. The main chemical components are CaO, SiO_2_, Al_2_O_3_, etc. The main mineral components are calcite and cacoclasite, and there are obvious slow scattering peaks between 20° and 40° (Figure 1), indicating that the slag is mainly in the amorphous glass phase and has high pozzolanic activity. The cement is a PI42.5 benchmark cement, produced by Fushun Aosaier Technology Co., Ltd., Fushun City, China.

Figure 2 is the SEM image of phosphogypsum, red mud, and slag. It can be seen from the figure that the phosphogypsum has a sheetlike structure (Figure 2a), the red mud has an irregular shape and a rough surface (Figure 2b), and the slag has irregular particles (Figure 2c).

Figure 3 shows the particle size distribution of phosphogypsum, red mud, slag, and cement. The DV (50) values of phosphogypsum, red mud, slag, and cement are 37.07, 11.21, 8.97, and 6.63 µm.

### 2.2. Mixture Preparation

The mixture proportion design of the composite cementitious material is shown in Table 2, in which the cement is an external admixture, and the water-solid ratio of all samples is 0.5. Here, R stands for red mud; S stands for slag; and P stands for phosphogypsum. Firstly, the amount of slag is determined. The ratio of slag increased from 0% (RS-0) to 50% (RS-5), and the ratio of red mud decreased from 100% to 50%. By comparing RS-0 and RS-1, RS-0 and RS-2, RS-0 and RS-3, RS-0 and RS-4, and RS-0 and RS-5, the effect of replacing red mud with slag on the mechanical properties of the composite cementitious materials was clarified. It has been reported that gypsum has a sulfate-stimulated effect similar to that of Na_2_SO_4_ [21]. Thus, in order to fully activate the potential hydration activity of red mud and slag, according to the influence of the replacement of red mud with slag on the mechanical properties of the composite cementitious material, the mixing amount of red mud and slag is determined to be 1:1, and phosphogypsum is used to replace slag. By comparing PRS-0 and PRS-1, PRS-0 and PRS-2, PRS-0 and PRS-3, PRS-0 and PRS-4, and PRS-0 and PRS-5, the effects of phosphogypsum replacement slag on the mechanical properties of composite cementitious materials was clarified.

We weighed the raw materials according to the mixture proportion in Table 2, poured them into the mixer, and stirred at a low speed for 3 min to evenly mix the raw materials. Water was added and stirred at a low speed for 2 min, and then at a high speed for 3 min. After stirring, the mixture slurry was poured into a 40 mm × 40 mm × 160 mm prismatic model. In order to eliminate the influence of air bubbles, vibration molding was adopted, and the mixture slurry was poured into the trial mold twice; each vibration lasted 30 s for a total of 60 s. After molding, it was left indoors for 24 h for disassembly and placed in a standard room for 7 days and 28 days for curing and strength testing.

### 2.3. Testing Methods

The strengths of the specimens were tested according to GB/T 17671-1999 [24]. The flexural and compressive strengths were tested with a digital display Model dkz-6000 electric flexure testing device (Made by Wuxi Jianyi Instrument Machinery Co., Ltd., Wuxi City, China) and a microcomputer-controlled electro-hydraulic cement press (YAW-300B, Made by Jinan Shijin Group Co., Ltd., Jinan City, China). The sample size was 40 mm × 40 mm × 160 mm, the flexural strength test was performed under the conditions of center line load, supporting span of 100 mm and acceleration rate of 50 N/s, and the test results gave three average values. For the compressive strength test, the loading rate was set to 500 N/s, and the test result is the average value of 6 repetitions.

### 2.4. Characterization Methods

X-ray diffraction analysis (XRD) was carried out to identify the phases inside the raw materials and the phase of the hydration product of the composite cementitious material. An Ultima IV X-ray Cu Kα radiation diffractometer (Rigaku, Tokyo, Japan) was used to collect X-ray diffractograms. We used copper target radiation and continuous scanning, with a wavelength of 0.15406 nm, a voltage of 40 KV and a current of 40 mA, 5 °/min.

Thermogravimetric analysis (TG-DTG) was performed to determine the effect of the slag and phosphogypsum contents on the hydration products of composite cementitious materials. A STA 8000 thermogravimetric analyzer (PerkinElmer, Waltham, MA, USA) was used to record the loss of weight of the samples during the thermal treatment. The TG analysis was performed under inert nitrogen protection conditions, with a heating rate of 15 °C/min, a temperature range of 45–900 °C, and a N_2_ flow rate of 50 mL/min.

Scanning electron microscopy (SEM) was used to study the microstructural hydration products and micromorphological properties of the composite cementitious materials. Prior to observation, the samples were dried and gold-plated in an ion sputter coater for 2 min. The microstructure was examined under a JSM-7800F scanning electron microscope (Jeol, Tokyo, Japan) with an accelerating voltage of 2–15 kV.

## 3. Results and Discussion

### 3.1. Determination of Slag Content and Mechanism Analysis

The effects of slag content on the strength of the composite cementitious material are shown in Figure 4. It can be seen that the compressive strength of RS-0 is lower, and there is no obvious change with the increase in curing age. Due to the low hydration activity of red mud, we can see fewer substances involved in the hydration reaction, fewer hydration products, and lower compressive strength and flexural strength [25]. Compared with the samples without slag (RS-0), the samples containing slag (RS-1, RS-2, RS-3, RS-4, and RS-5) showed higher compressive strength and flexural strength, and this increased gradually with the increase in curing age and slag content. Among them, the compressive strength of RS-3 at 7 days and 28 days increased from the 3.5 MPa and 3.7 MPa values of RS-0 to 10.4 MPa and 13 MPa, which are improvements of 191% and 251%, respectively (Figure 5). Subsequently, the slag content continued to increase, and the 7- and 28-day compressive strength of the composite cementitious material showed slow growth, with said values of RS-4 being basically the same as those of RS-3 (Figure 4 and Figure 5). When the slag content was increased to 50% (RS-5), the 7- and 28-day strengths reached the maximum. Among them, the 7- and 28-day compressive strengths of RS-0 increased from 3.5 MPa and 3.7 MPa to 11.2 MPa and 14.0 MPa—220% and 278% improvements, respectively. However, improvements of only 23% and 27% were seen for RS-3 (Figure 5). This shows that the addition of slag can effectively improve the strength of the composite cementitious material. The active aluminum and silicon components in the composite cementitious material increased with the increase in slag, the aluminum and silicon participating in the hydration reaction’s increase, and the strength of the composite cementitious material was improved. However, when the amount of slag increased, the amount of red mud remained relatively small, the alkalinity of the system was weakened, and the strength growth of the composite cementitious material gradually slowed.

Although red mud contains many aluminosilicates, its activity is much lower than that of slag; the improvement in the strength of composite cementitious materials is mainly controlled by the hydration of slag. Red mud mainly provides the action of an alkali, thereby activating the hydration of the slag. A proper alkaline environment is crucial for the hydration of slag, and the ratio of slag to red mud determines the alkalinity, which affects the strength development. The experiment results show using 1:1 red mud: slag yields a composite cementitious material with the highest strength. 

In general, there is a linear relationship between compressive strength and flexural strength [26]. Furthermore, Figure 6 shows the relationship between compressive strength and flexural strength. They have a linear relationship, but its R^2^ value is low (R^2^ = 0.7071), which may be caused by the experimental error in the flexural strength test.

Figure 7 shows the XRD patterns of composite cementitious materials RS-0, RS-3, and RS-5. The main crystalline substances are sodium zeolite, cancrinite, dicalcium silicate, tricalcium aluminate, hematite, goethite, boehmite, gibbsite, and calcite. It can be seen from the figure that no new characteristic peaks appear when the slag is added, which indicates that the addition of the slag will not change the type of crystal substance of the red mud-based cementitious material. The improvement in the strength of the composite cementitious material may be mainly attributed to the amorphous C-(A)-S-H produced by the hydration of the slag and red mud. Research by Si [27] shows that the CaO released from the slag can undergo a Na^+^ ion replacement reaction with the cancrinite in red mud to improve the alkalinity of the system. In an alkaline environment, the aluminosilicate glass of the slag and red mud dissolves and react with Ca(OH)_2_ in the solution. However, the reduction in Ca(OH)_2_ can promote the hydration of C_2_S and C_3_A [28]. Therefore, with the increased content of slag, the strength of the composite cementitious material increased too. The amorphous hydration product was analyzed by TG-DTG and SEM. With the increase in slag content, the characteristic peaks of the crystal material gradually weakened, and this is mainly attributable to the reduction in red mud content. It is well known that cement hydration can generate large amounts of Ca(OH)_2_, but Ca(OH)_2_ was not found in the XRD pattern. This means that the Ca(OH)_2_ has been consumed by the pozzolanic activity of slag and red mud [29,30].

Numerous studies have shown [23,31] that the amount of Ca(OH)_2_ and C-(A)-S-H in the composite cementitious material can indirectly indicate the strength of the hydration product. The lower the Ca(OH)_2_ content, the higher the strength of the hydration product. Figure 8 shows the TG-DTG curves of the composite cementitious materials RS-0 and RS-5 hydrated for 28 days. The positions of the endothermic peaks of the two composite cementitious materials are basically the same, indicating that the replacement of red mud with slag will not change the hydration products of the red mud-based composite cementitious materials.

The stage before 100 °C is the loss of chemically bound water. The stage around 80 °C may be primarily caused by the evaporation of the bound water in sodium zeolite and cancrinite. At this time, the endothermic peak of RS-0 was stronger than that of RS-5, indicating that RS-0 contains more sodium zeolite and cancrinite. This is consistent with the XRD results. The 100–200 °C stage was mainly influenced by the weight loss of hydration products, and that around 140 °C was mainly determined by the dehydration of C-(A)-S-H [23]. In this range of temperature, the endothermic peak of RS-5 was stronger than that of RS-0, indicating that the R-5 contains more C-(A)-S-H gelling components. Thus, RS-5′s strength is much greater than that of R-0. The weight loss between 250 and 300 °C can be attributed to the loss of crystallization water from the hydroxides [19]; at 270 °C, goethite and gibbsite may lose water and convert to hematite and boehmite [32,33], respectively. The endothermic peak of RS-0 was stronger than that of RS-5, indicating that the content of goethite and gibbsite in RS-0 is greater, which is consistent with the XRD results. The weight loss at 300–400 °C was mainly due to the dehydroxylation of the Al-O layer in tricalcium aluminate and cancrinite [34]. The endothermic peak of RS-0 was stronger than that of RS-5 at 340 °C, indicating that the content of tricalcium aluminate and cancrinite in the RS-0 sample is greater than that of RS-5, which is consistent with the XRD results. Generally, there was no Ca(OH)_2_ peak at 450 °C, indicating that the hydration reaction of the composite cementitious material was sufficient, and the Ca(OH)_2_ had been consumed at 28 days, which is consistent with the XRD results. The weight loss of the sample around 550 °C may be related to the transformation of boehmite into corundum (Al_2_O_3_), and the destruction and transformation of the sodium zeolite lattice into the amorphous state [35]. The endothermic peak of RS-0 was stronger than that of RS-5, indicating that RS-0 contains more boehmite and sodium zeolite, which is consistent with the XRD results. The endothermic peak around 650–750 °C was mainly caused by the decomposition of carbonate into calcite and cancrinite, or the dehydroxylation of the cancrinite and dicalcium silicate Si-O layer [34,36]. Figure 8a shows that the weight loss rate of RS-5 (16.85) was much greater than that of RS-0 (13.74). This illustrates that the higher the content of bound water in RS-5, the higher the degree of hydration, and the greater the presence of C-(A)-S-H hydration products [37].

Figure 9 shows the microstructures of RS-0, RS-3, and RS-5 hydrated for 28 days. The results show that the hydration products of the composite cementitious material are mainly reticulated C-(A)-S-H. There are basically unreacted red mud particles in RS-0, indicating that there are few hydration products, and most of the red mud particles only play the role of micro-aggregate filling in the composite cementitious material, contributing little to the strength, which verifies the conclusion that red mud has low hydration activity, fewer hydration products, and low strength. Compared with RS-0, the red mud particles in RS-3 are obviously reduced, and the reticulated C-(A)-S-H was more obvious. This shows that under the action of the OH^−^ released from red mud and cement, the activity of the SiO_2_ and Al_2_O_3_ in the slag increases, and this participates in the hydration reaction to generate more reticulated C-(A)-S-H. The C-(A)-S-H encapsulates red mud particles to form a hardened structure with high strength. Compared with RS-0 and RS-3, the reticulated hydration product C-(A)-S-H in RS−5 is particularly obvious and contains a small amount of unreacted red mud particles. The structure of the hardened body was more uniform and compact. This indicates that the greater the amount of slag admixture used, the greater the amount of hydrated C-(A)-S-H gelation that is generated, which is consistent with the TG-DTG results.

### 3.2. Influence of Phosphogypsum on the Strength of Composite Cementitious Materials and Mechanism Analysis

The effect of phosphogypsum content on the strength of the composite cementitious material is shown in Figure 10. It can be seen that PR-1, PRS-2, and PRS-3 show greater strength than PRS-0. This shows that the proper amount of phosphogypsum can effectively improve the strength of the composite cementitious material. With the increase in phosphogypsum content, the strength of the composite cementitious material increased first, and then decreased. Among them, the 7- and 28-day compressive strengths of PRS-1 increased from 11.2 MPa and 14.0 MPa (the result for PRS-0) to 19.3 MPa and 25.0 MPa, respectively, and reached the maximum, manifesting increases of 72% and 79%, respectively (Figure 11). When the phosphogypsum increased to 20% (PRS-2), the 7- and 28-day strengths began to decrease slightly, of which the compressive strengths were 17.8 MPa and 24 MPa, respectively, which increased by 59% and 71% compared with PRS−0 (Figure 11). When the phosphogypsum was increased to 30% (PRS-3), the 7-day and 28-day strength decreased significantly; the 7-day compressive strength was 13 MPa and the 28-day compressive strength was 20 MPa, which are increases of only 16% and 43% compared with PRS-0 (Figure 11). Numerous studies have shown [12,21,38] that gypsum can accelerate the pozzolanic reaction of slag, fly ash, and other materials, generate more C-S-H and an ettringite-reinforced matrix, and improve the strength. Usually, when the content of phosphogypsum is 15–20%, the composite cementitious material has the best strength [9].

Generally speaking, due to the low crystal strength of phosphogypsum and its limited participation in the reaction, when it is compounded with other active substances to prepare cementitious materials and the dosage is excessive, the hydration product content will be reduced, and the strength of the composite cementitious materials will be drastically reduced. Compared with PRS-0, the 7- and 28-day strengths of PRS-4 decreased sharply; the 7-day strength was lower than that of PRS-0, having decreased by 15%, and the 28-day compressive strength only increased by 18% (Figure 11). When the phosphogypsum completely replaced the slag (PRS-5), the 7- and 28-day strengths were lower than those of PRS-0, and the shrinkage phenomenon occurred with curing age. In addition, when curing for 28 days, the sample underwent serious volume expansion damage, which made it impossible to test its flexural strength. This may be related to ettringite expansion damage. Figure 12 shows the relationship between compressive strength and flexural strength. It can be seen that they have an obvious linear relationship, and the R^2^ value is 0.80923.

The XRD patterns of the composite cementitious materials with different phosphogypsum contents are shown in Figure 13. The main crystal substances of the composite cementitious material are ettringite, calcium sulfate dihydrate, and hematite. Compared with PRS-0, PRS-1, PRS-2, PRS-3, PRS-4, and PRS-5 have obvious characteristic peaks of ettringite and calcium sulfate dihydrate. This is due to the increased activity of SiO_2_ and Al_2_O_3_ in slag and red mud under the action of OH^−^, and they react with the cement hydration product Ca(OH)_2_ to generate C-(A)-S-H. A large amount of C-A-H can simultaneously react with calcium sulfate dihydrate in phosphogypsum to form ettringite [39]. Phosphogypsum can accelerate the activity of slag and red mud pozzolan to form ettringite. All the Ca(OH)_2_ was consumed during the hydration process, which is why Ca(OH)_2_ was not detected in the XRD results. The characteristic peaks of ettringite increased first and then decreased with the increase in phosphogypsum content, indicating that there was an optimal range of phosphogypsum content. With the increase in the phosphogypsum content, the characteristic peaks of calcium sulfate dihydrate gradually increased, which indicates that the phosphogypsum participating in the hydration reaction was limited. Most of them can only serve as the matrix of microaggregates-filled composite cementitious materials. However, the strength of the gypsum crystal itself is very low, and the excessive content can reduce the strength of the composite cementitious material.

It is precisely due to the formation of much ettringite that the dissolution of AlO^2^^−^ and Ca^2+^ in red mud and slag is promoted, resulting in the depolymerization of the vitreous structure. On the other hand, as regards the Al_2_O_3_ in the composite cementitious material, except for the formation of ettringite, most of the rest are network formers. In addition, there are also a few intermediates whose bond energy is one-third to half of that of the network former, SO_4_^2^^−^ is easily adsorbed onto these particles to erode the vitreous body, causing it to disintegrate, while Al-O breaks, and this also promotes the breaking of Si-O [27]. At the same time, the presence of SO_4_^2^^−^ can change the water permeability of C-S-H, thereby accelerating the formation of C-S-H and improving the strength of the composite cementitious material [27]. The characteristic peak of ettringite in PRS-3 is the strongest, indicating that phosphogypsum can greatly accelerate the dissolution of Al^3+^ in slag and red mud and react with calcium sulfate dihydrate to form ettringite within a certain dosage range. However, the strength of PRS-3 has been greatly reduced, which may be attributed to the large amount of ettringite formed in the capillary pores of the composite cementitious material, whose size is larger than that of the capillary pores; this material is prone to expansive destruction, which reduces the strength [40].

When the content of phosphogypsum reaches 40%, the characteristic peak intensity of ettringite decreases. The content of phosphogypsum is increased, and the content of slag is relatively reduced, which means that the content of phosphoric acid in phosphogypsum is increased, and the content of Al_2_O_3_ and SiO_2_ is reduced. However, the reduction in Al_2_O_3_ and SiO_2_ content inevitably reduces the formation of ettringite and C-(A)-S-H. In addition, the acidic impurities in phosphogypsum can also reduce the alkalinity of the system, hinder the formation of ettringite and C-(A)-S-H and the crystal structure of the crude hydration product, and reduce the strength of the composite cementitious material [10,41]. Therefore, PRS-4 and PRS-5 show extremely poor mechanical properties. Compared with PRS-0, PRS-5 maintains fairly strong characteristic peaks of ettringite, indicating that phosphogypsum can stimulate the pozzolanic activity of red mud and participate in the hydration reaction.

Red mud contains a large amount of iron phase (Table 1), mainly in the form of hematite. The effect of iron on geopolymerization reactions has attracted the attention of researchers. It has been reported that any active iron would rapidly reprecipitate in the form of hydroxide or oxy-hydroxide phases, during fly ash geopolymerization, thereby consuming OH^−^ ions from the solution phase, slowing the dissolution of the remaining fly ash particles, and providing a nucleation site [16]. Daux et al. [42] found that a large amount of netlike Fe^3+^ was dissolved from basalt glass in a weakly alkaline solution, and the dissolved Fe was reprecipitated faster than Si and Al. The iron phase in the red mud underwent reprecipitation during the high-alkali refining of alumina [16]. Therefore, Fe in red mud is rarely present in the glass body, such as in the slag forming a network. The results in Figure 8 show that after the hydration reaction of the composite cementitious material, the hematite remains basically unchanged, indicating that Fe has little effect on the composite cementitious material.

Figure 14 shows the TG-DTG curves of the composite cementitious materials with different phosphogypsum contents after curing for 28 days. There are obvious endothermic peaks around 100 °C and 130 °C. Numerous studies have shown [43,44,45,46] that the water loss peaks of C-(A)-S-H and ettringite are around 100 °C. With the increase in phosphogypsum content, the overlapping peaks of C-(A)-S-H and ettringite increase first and then weaken. When the content of phosphogypsum increases to 30%, the peak value becomes the most intense, and then gradually weakens, which is consistent with the XRD results. The addition of phosphogypsum provides a large amount of SO_4_^2^^−^ for the formation of ettringite, and within a certain range, the higher the SO_4_^2^^−^ content, the greater the amount of ettringite produced. In addition, SO_4_^2^^−^ can change the charge distribution of the system, thereby accelerating the dissolution of Si^4+^ and Al^3+^ in red mud and slag to generate more ettringite and C-(A)-S-H [21]; thereby, the strength of the composite cementitious material is significantly improved. With the increase in phosphogypsum, the contents of SiO_2_ and Al_2_O_3_ in the system decrease, the ettringite and C-(A)-S-H decrease accordingly, the endothermic peak around 100 °C weakens, and the strength of the composite material decreases sharply. The peak at 130 °C is caused by the dehydration of calcium sulfate dihydrate [21,47], and with the increase in phosphogypsum content, its peak value gradually strengthens. The XRD results have been verified: a limited amount of the phosphogypsum takes part in the hydration reaction, most of which only acts as an aggregate filler. Phosphogypsum crystals have extremely low strength, and as micro–aggregate filling, contribute little to the strength of composite cementitious materials. Therefore, when the content of phosphogypsum increases to 40%, the strength of the composite cementitious material decreases sharply. The characteristic peak of Ca(OH)_2_ that usually appears around 450 °C does not exist, indicating that Ca(OH)_2_ has been consumed in the hydration process of the composite cementitious material, which is consistent with the XRD results.

Generally speaking, the more chemically bound the water is, the higher the degree of hydration of the composite material, and the more hydration products and the higher the strength of the hardened body [37]. Figure 15 shows the relationship between the 28-day compressive strength and chemically bound water. The result shows that they have an obvious linear relationship, and the R^2^ value is 0.8212. The order of weight loss rate is PRS-1 (15.2) > PRS-2 (15.03) > PRS-3 (13.74) > PRS-0 (12.93) > PRS-5 (9.13) > PRS-4 (8.42), which is basically consistent with the strength development. Among these, the weight loss rate of PRS-5 is higher than that of PRS-4, but its strength is lower than that of PRS-4. This may be related to the water loss of PRS-5, which contains more unreacted calcium sulfate dihydrate.

Figure 16 shows the SEM image of the composite cementitious material (28 days) with different phosphogypsum contents. It can be seen from the figure that the hydration products of the composite cementitious material are mainly hydrated C-(A)-S-H, ettringite. Compared with the sample without phosphogypsum (PRS-0), a large amount of ettringite and unreacted calcium sulfate dihydrate was added to the sample containing phosphogypsum (PRS-1, PRS-2, PRS-3, PRS-4, and PRS-5). Due primarily to the presence of SO_4_^2−^, this material easily reacts with Al^3+^, Ca^2+^, OH^−^ to form ettringite. It is once again verified that phosphogypsum can accelerate the pozzolanic reaction of red mud and slag. The PRS-0 only contains reticulated C-(A)-S-H and red mud particles, and the density of the matrix is poor. On the other hand, PRS-1 and PRS-2 contain more reticular C-(A)-S-H and acicular ettringite, and a large amount of C-(A)-S-H and ettringite interweave with each other to enclose the red mud particles and unreacted phosphogypsum particles in a more uniform and compact matrix. Numerous studies have shown [21,48,49] that SO_4_^2−^ can not only accelerate the formation of ettringite, but also change the charge of the system to accelerate the dissolution rate of Si^4+^ and Al^3+^ in red mud and slag, thereby forming more C-(A)-S-H to improve the strength of composite cementitious materials. Therefore, PRS-1 and PRS-2 have greater strength, which also proves the conclusions of XRD and TG-DTG. When the content of phosphogypsum was increased to 30%, many ettringite crystals with a disorderly distribution appeared in the composite cementitious material, and larger pores appeared, which is consistent with the detection results of XRD and TG-DTG. The concentration of SO_4_^2−^ increases with the increase in phosphogypsum content, which provides the conditions for the formation of a large amount of ettringite, and a large amount of ettringite is prone to expansion damage. At the same time, the reduction in the slag can also reduce the formation of C-(A)-S-H. Therefore, the strength of PRS-3 is greatly reduced. When the phosphogypsum increased to 40% (PRS-4), the composite cementitious material ettringite decreased, and unreacted calcium sulfate dehydrate was apparent, which is consistent with the detection results of XRD and TG-DTG. The presence of a large amount of phosphogypsum greatly reduces the content of slag, resulting in a decrease in the content of a large amount of active SiO_2_ and Al_2_O_3_; furthermore, the hydration products of the composite cementitious material decrease, and thus the strength drops sharply. When the phosphogypsum is increased to 50% (PRS-5), the amount of slag is zero, but a large amount of ettringite is still densely accumulated on the surfaces of other hydration products or unreacted raw material particles. In addition, significant amounts of unreacted calcium sulfate dihydrate were also found. The conclusions of XRD and TG-DTG have been proven-phosphogypsum can improve the hydration activity of red mud. When PRS-5 is cured for 28 days, it cannot be put into a test mold to test its flexural strength due to expansion. This can be attributed to the expansion of ettringite with the presence of a small amount of C-(A)-S-H. The expansion of ettringite and the presence of many low-strength gypsum crystals sharply reduced the overall strength of the sample. Therefore, the strength of PRS-5 is much lower than that of PRS-0, and it drops further with curing age.

## 4. Conclusions

In this study, phosphogypsum-red mud-slag composite cementitious materials were prepared using phosphogypsum, red mud, and slag as the main raw materials, and extra cement was added as the curing agent. We assessed the law determining the relationship between the strengths of phosphogypsum-red mud-slag composite cementitious materials. The hydration products and microstructure of the composite cementitious materials were analyzed via XRD, TG-DTG, and SEM and the composite cementitious materials were revealed. As regards the strength enhancement mechanism, the following conclusions have been drawn.

The slag has a significant strengthening effect on the composite cementitious material. When the slag content is 0–50%, the strength of the composite cementitious material increases with the increase in the slag content. When the slag content increases to 50%, the composite cementitious material shows the highest strength, and its 28-day compressive strength can reach 14.0 MPa.Phosphogypsum has a significant effect on the strength development of the composite cementitious material. When the content of phosphogypsum is 0–50%, the strength of the composite cementitious material increases first and then decreases. When the content of phosphogypsum is 10%, the composite cementitious material has the highest strength, and its 28-day compressive strength reaches 25.0 MPa.Phosphogypsum can improve the hydration activity of slag and red mud, mainly because the SO_4_^2−^ released in phosphogypsum can react with the hydration product C-A-H to produce ettringite and promote the dissolution of Al^3+^, Ca^2+^, and Si^4+^ in slag and red mud, generating more C-(A)-S-H and ettringite, which improves the strength of the composite cementitious material.The hydration products of phosphogypsum-red mud-slag composite cementitious materials are mainly C-(A)-S-H and ettringite, of which C-(A)-S-H binds ettringite and unreacted particles into a dense structure, thereby improving the strength of the composite cementitious material.

## Figures and Tables

**Figure 1 materials-15-06096-f001:**
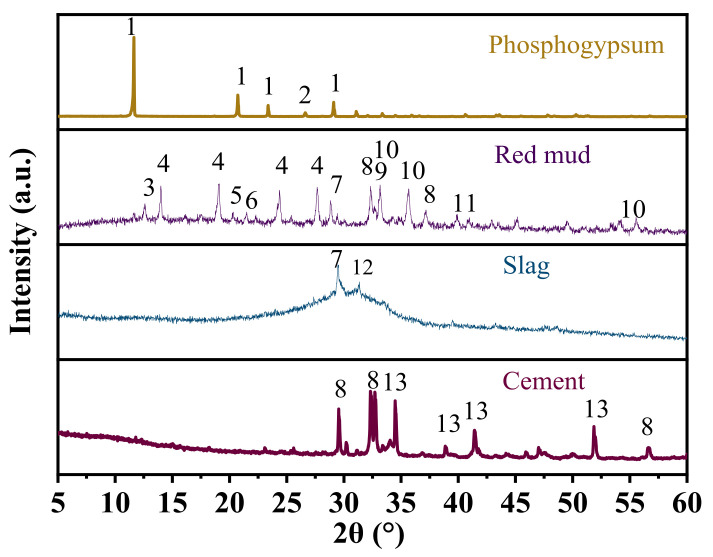
XRD patterns of main materials (1: calcium sulfate dihydrate; 2: quartz; 3: sodium zeolite; 4: cancrinite; 5: goethite; 6: boehmite; 7: calcite; 8: dicalcium silicate; 9: tricalcium aluminate; 10: hematite; 11: gibbsite; 12: cacoclasite; 13: tricalcium silicate).

**Figure 2 materials-15-06096-f002:**
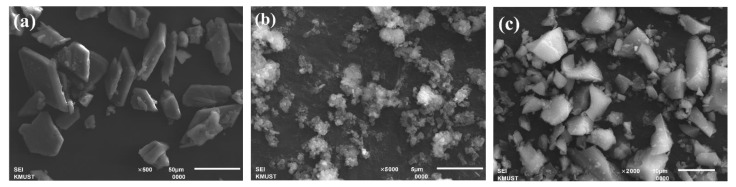
SEM images of (**a**) phosphogypsum; (**b**) red mud; and (**c**) slag.

**Figure 3 materials-15-06096-f003:**
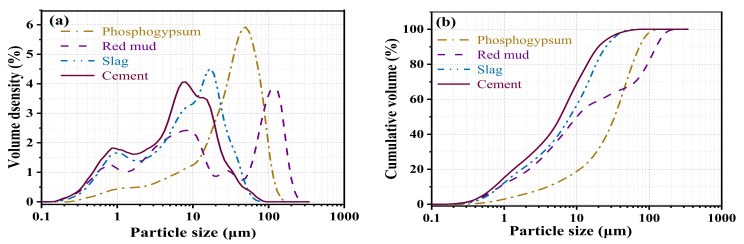
Particle size distributions of phosphogypsum, red mud, slag, and cement ((**a**): volume density; (**b**): cumulative volume).

**Figure 4 materials-15-06096-f004:**
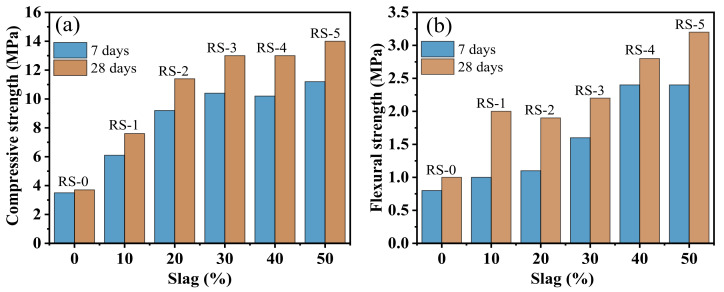
(**a**) Compressive strength and (**b**) flexural strength of composite cementitious material at 7 days and 28 days with different slag contents.

**Figure 5 materials-15-06096-f005:**
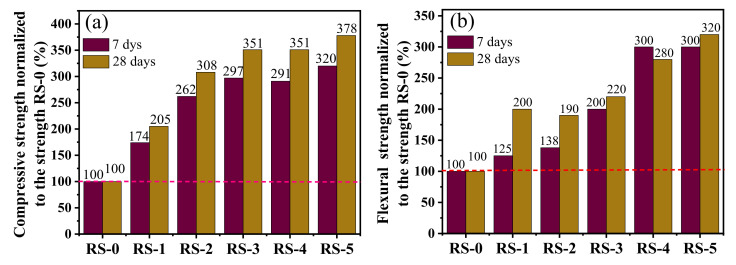
(**a**) Compressive strengths normalized to the strength RS-0; (**b**) flexural strengths normalized to the strength of RS-0.

**Figure 6 materials-15-06096-f006:**
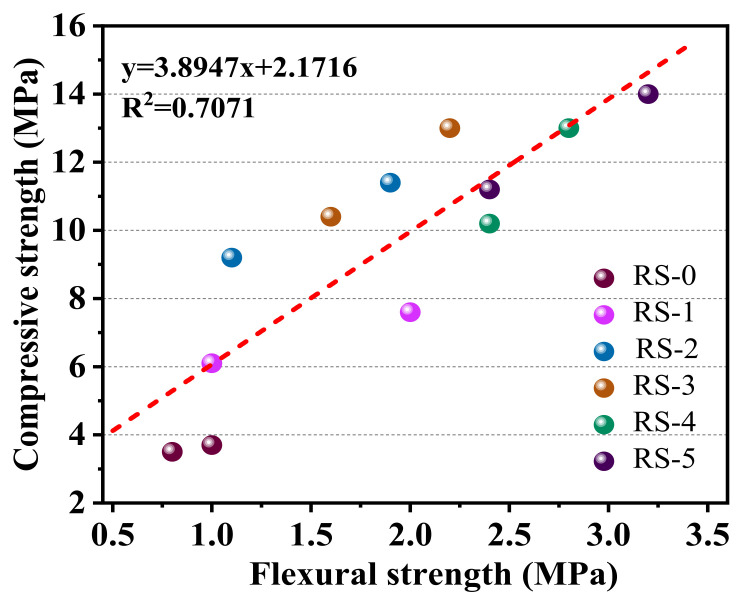
The relationship between compressive strength and flexural strength of composite cementitious materials with different slag contents.

**Figure 7 materials-15-06096-f007:**
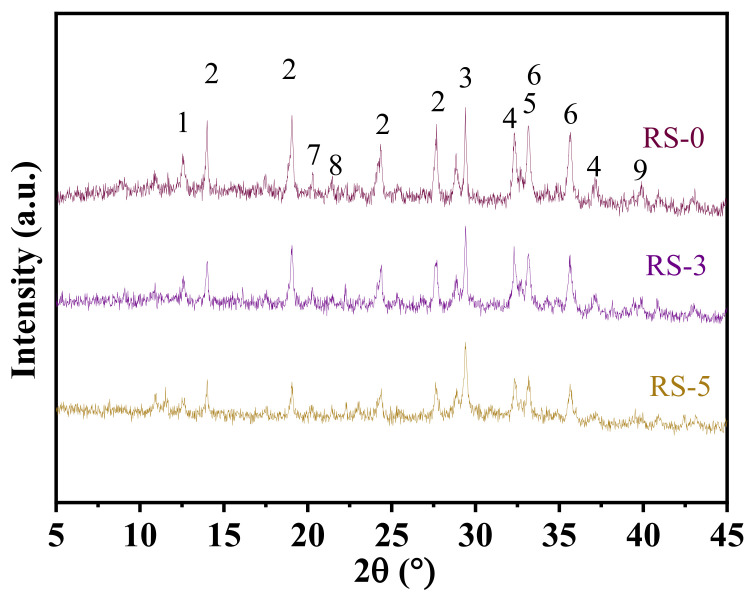
XRD patterns of composite cementitious material at 28 days with different slag contents (1: sodium zeolite; 2: cancrinite; 3: calcite; 4: dicalcium silicate; 5: tricalcium aluminate; 6: hematite; 7: goethite; 8: boehmite; 9: gibbsite).

**Figure 8 materials-15-06096-f008:**
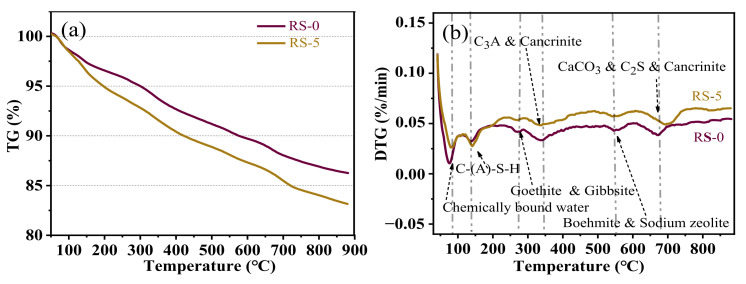
(**a**) The TG and (**b**) DTG curves of composite cementitious materials with different slag contents at 28 days.

**Figure 9 materials-15-06096-f009:**
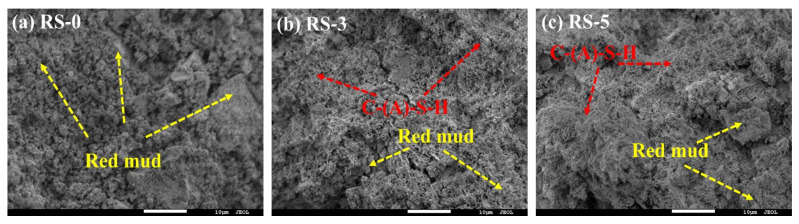
SEM images of (**a**) RS-0; (**b**) RS-3; and (**c**) RS-5 at the curing age of 28 days.

**Figure 10 materials-15-06096-f010:**
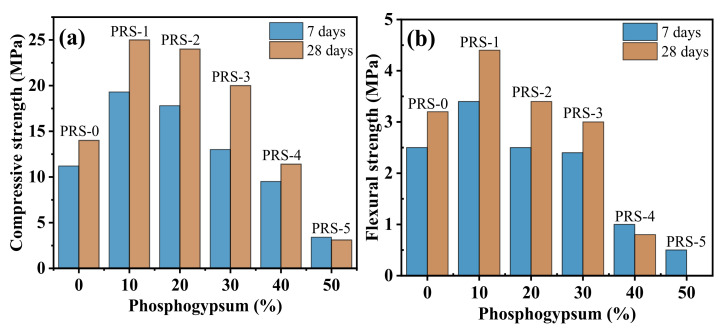
(**a**) Compressive strength and (**b**) flexural strength of composite cementitious material at 7 days and 28 days with different phosphogypsum contents.

**Figure 11 materials-15-06096-f011:**
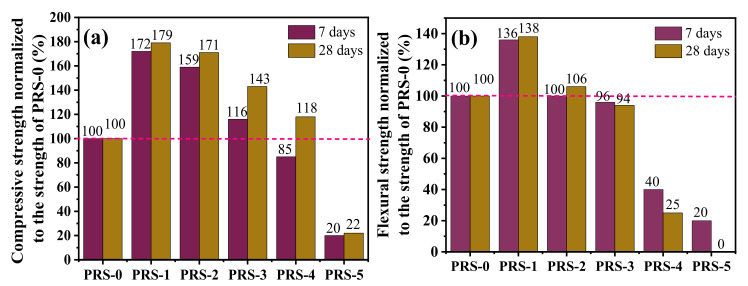
(**a**) Compressive strengths normalized to the strength of RS-0; (**b**) flexural strengths normalized to the strength of RS-0.

**Figure 12 materials-15-06096-f012:**
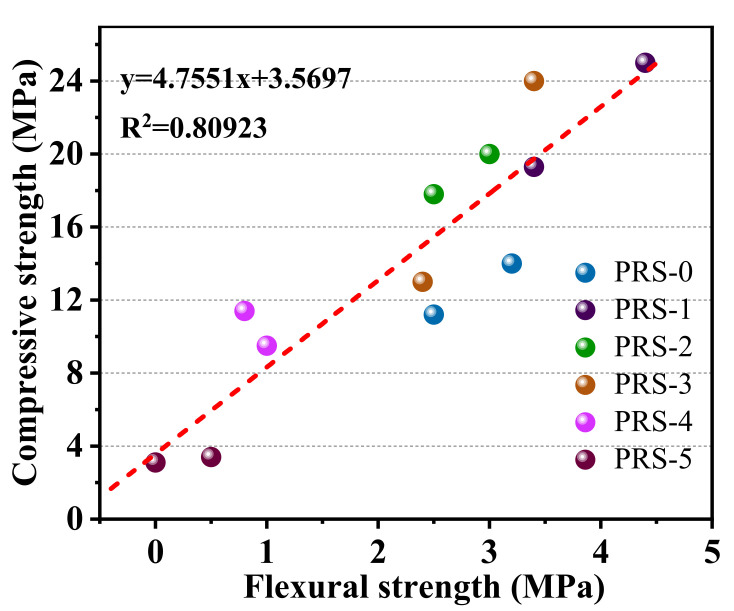
The relationship between the compressive strength and flexural strength of composite cementitious materials with different phosphogypsum contents.

**Figure 13 materials-15-06096-f013:**
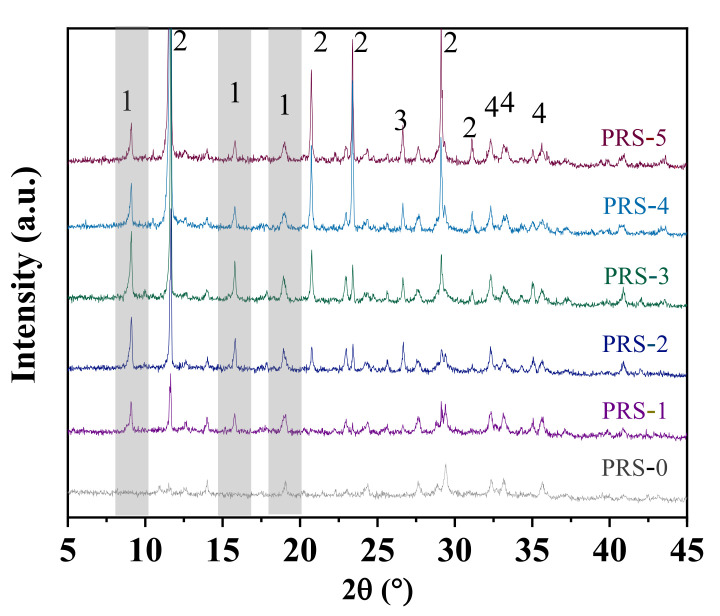
XRD patterns of the composite cementitious material at 28 days with different phosphogypsum contents (1: ettringite; 2: calcium sulfate dihydrate; 3: SiO_2_; 4: hematite).

**Figure 14 materials-15-06096-f014:**
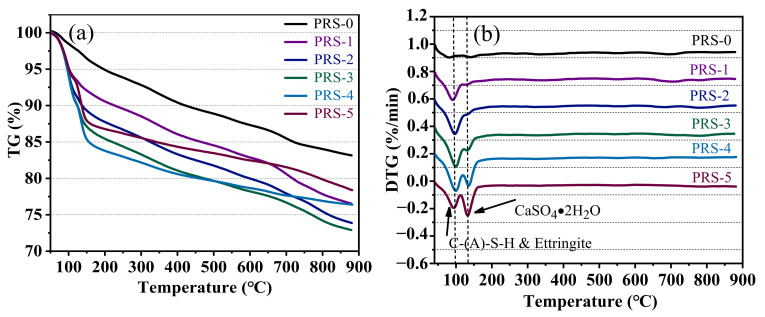
(**a**) TG and (**b**) DTG curves of composite cementitious materials with different phosphogypsum contents (28 days).

**Figure 15 materials-15-06096-f015:**
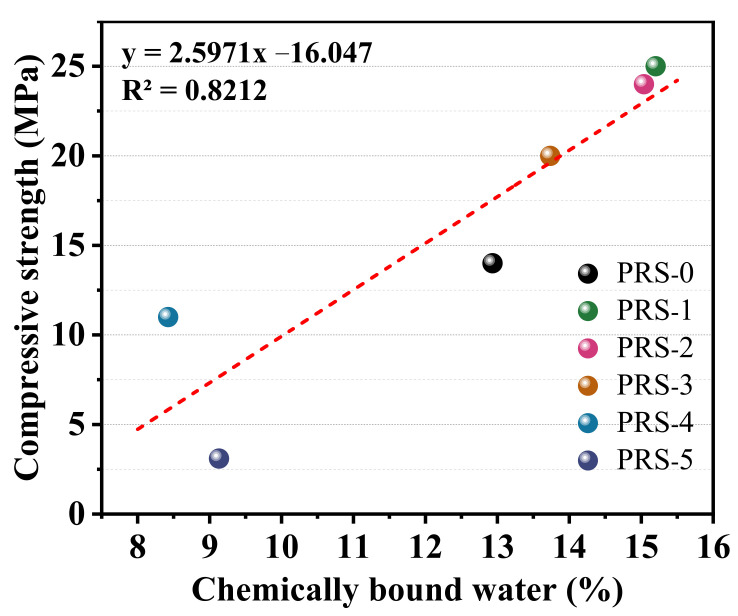
The relationship between chemically bound water and the compressive strength of 28-day composite cementitious materials (160–900 °C).

**Figure 16 materials-15-06096-f016:**
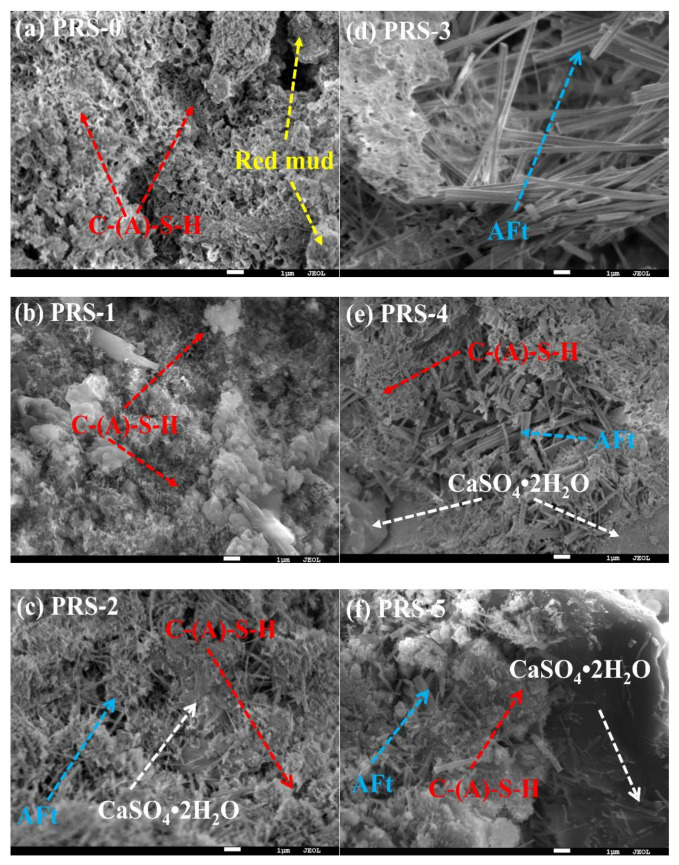
SEM images of (**a**–**f**) at the curing age of 28 days.

**Table 1 materials-15-06096-t001:** Chemical compositions of raw materials (wt. %).

Raw Materials	SO_3_	CaO	SiO_2_	Al_2_O_3_	Fe_2_O_3_	Na_2_O	MgO	TiO_2_	F	P_2_O_5_	Other	Total
Phosphogypsum	47.59	37.93	11.88	0.62	0.14	0.31	0.12	-	0.29	0.67	0.45	100
Red mud	0.71	12.76	17.12	19.23	30.74	11.83	0.31	5.53	-	-	1.77	100
Slag	1.98	40.68	34.48	13.43	0.29	-	5.7	2.57	-	-	0.87	100
Cement	2.21	63.43	20.84	4.68	3.62	0.56	3.29	-	-	-	1.37	100

**Table 2 materials-15-06096-t002:** Mixture proportion (wt. %).

No.	Red Mud	Slag	Phosphogypsum	Cement
R-0	100	0	0	10
RS-1	90	10	0	10
RS-2	80	20	0	10
RS-3	70	30	0	10
RS-4	60	40	0	10
RS-5	50	50	0	10
PRS-0	50	50	0	10
PRS-1	50	40	10	10
PRS-2	50	30	20	10
PRS-3	50	20	30	10
PRS-4	50	10	40	10
PRS-5	50	0	50	10

## Data Availability

Not applicable.

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
