# Peer review of "Microstructure and Key Properties of Phosphogypsum-Red Mud-Slag Composite Cementitious Materials"

_materials, 2022, doi:10.3390/ma15176096_

Round 1
Reviewer 1 Report
The use of phosphogypsum, red mud and slag powder to prepare composite cementitious materials has positive significance for energy saving and environmental protection. Compared with the existing research, the main contribution of this paper is to prepare cementitious materials based on phosphogypsum and red mud, two common solid waste materials, and systematically study the influence of the dosage of phosphogypsum and slag powder on the strength, hydration products and microstructure of composite cementitious materials. The article has a clear goal and clear thinking, which is of great significance to the resource utilization of phosphogypsum and red mud. However, the following issues need to be modified and supplemented.
1. Line 36, please correct the writing format of calcium sulfate dihydrate (CaSO4·2H2O).
2. In line 330, "Figure 7" does not correspond to the figure number, so change Figure 7 to Figure 8.
3. The line spacing between lines 345-351 is obviously inconsistent with the full text. Please check whether the fonts "RS−0", "RS−5", "RS0" and "Al−O" are wrong.
4. The font of "C−S−H" connecting symbol in line 502 is wrong, please modify it. The same problem occurred with "PRS−1" and "PRS−2" in lines 546-547. Please revise it. The "C−A−H" in line 594 has a similar problem. Please revise it.
5. As we all know, the main component of phosphogypsum is calcium sulfate dihydrate, but it also contains a lot of impurities. Using untreated phosphogypsum as the main raw material to prepare composite cementitious materials will inevitably introduce a lot of impurities. Please elaborate the influence of phosphogypsum impurities on the properties of the composite cementitious materials.
6. Red mud contains a lot of heavy metals and radioactive elements, which will have a serious negative impact on the ecological environment in engineering application. It is necessary to further evaluate the impact of heavy metals and radioactive elements on the performance and environment of composite cementitious materials.
7. In the process of alumina smelting, SiO2 and Al2O3 in red mud are converted into stable aluminosilicate, and its hydration reaction is extremely low, so it usually needs thermal activation to improve its activity. In this study, phosphogypsum is used to activate red mud. Please elaborate the mechanism of phosphogypsum activating red mud.
Author Response
Dear Reviewer:
We gratefully thank your constructive comments and helpful suggestions, which has significantly raised the quality of the manuscript and has enabled us to improve the manuscript. Each of your suggested revisions and comments was thoroughly considered and modified in the Revised Manuscript. In this file, the authors will answer the comments item by item.

Reviewer 2 Report
Thank you for submitting your paper. The work done here draws attention to a significant subject on Composite Cementitious Materials. I have found the paper to be interesting. However, several issues need to be addressed properly before the paper is being considered for publication. My comments including major and minor concerns are given below:
Please consider reviewing the abstract and highlight the novelty, major findings, and conclusions. I suggest reorganizing the abstract, highlighting the novelties introduced. The abstract should contain answers to the following questions:
What problem was studied and why is it important?
What methods were used?
What conclusions can be drawn from the results? (Please provide specific results and not generic ones).
The abstract must be improved. It should be expanded. Please use numbers or % terms to clearly shows us the results in your experimental work.
Please consider reporting on studies related to your work from mdpi journals.
An updated and complete literature review should be conducted. The author should write a well-analyzed Introduction in the standard length (shorten the current one a little bit).
Authors should avoid bulk citations. They should be either removed or individually discussed to give full credit to each of the papers.
The authors should mention any standards used for testing in the materials and methods section.
Table 1 needs a reference(s) if not measured by the authors
Why the authors choose those proportions in Table 2, is it according to industrial recommendation or randomly chosen by the authors
From figure 4 it is clear that addition of slag increases the compressive strength in a linear fashion but the same can not be said for flexural strength? How come the trends are different? Authors need to provide scientific explanation supported by references, also what about past studies, did they report similar trends?
Figure 6 R squared is very low, ideally it should be over 85% or even 95% to be considered reliable.
Lines 321-322 the authors need to support this claim with reference(s)
Author Response
Dear Reviewer:
We gratefully thank your constructive comments and helpful suggestions, which has significantly raised the quality of the manuscript and has enabled us to improve the manuscript. Each of your suggested revisions and comments was thoroughly considered and modified in the Revised Manuscript. And please see the attachment. In this file, the authors will answer the comments item by item.

Round 2
Reviewer 2 Report
All questions answered and paper can be accepted
Author Response
We gratefully thank your constructive comments.